# Lifestyle-Related Factors for Improving Diet Quality

**DOI:** 10.3390/nu17030448

**Published:** 2025-01-26

**Authors:** Derek Fischer, Ping Ouyang

**Affiliations:** Department of Family and Consumer Sciences, Western Michigan University, Kalamazoo, MI 49008, USA; derek.j.fischer@wmich.edu

**Keywords:** exercise, sleep, mindfulness, mindful eating, intuitive eating, social life, media use, smoking, alcohol

## Abstract

**Background/Objectives:** Diet quality is important for the prevention of non-communicable diseases (NCDs), which are now responsible for leading causes of death worldwide. Although health professionals often recommend diet improvement for the treatment and management of many NCDs, patients continually struggle to make dietary changes and maintain them long-term. This may be due to an interplay of many factors that affect dietary quality. This paper discusses some lifestyle-related factors that may offer additional points of intervention for health professionals to use to increase diet quality. **Methods:** This review examines the effects of exercise, sleep, mindfulness practice, meal socialization, social media use, and tobacco and alcohol use on diet quality. Studies examining the effects of these factors on diet were found by searching PubMed, CINAHL, and Google Scholar. **Results:** Evidence suggests that a positive relationship between exercise, adequate sleep, and all forms of mindfulness have positive effects on diet quality. Meal socialization’s effects on diet quality were nuanced. Individuals tend to eat similarly to those they share a meal with. However, food quantity intake may be affected by the physical characteristics of individuals with whom people share a meal. Social media use was found to have negative effects on diet quality in those who used it excessively and was found to promote disordered eating patterns. Both tobacco and alcohol use were found to have negative impacts on diet quality. **Conclusions:** Using these findings, health professionals may be able to educate people about lifestyle-related factors that can improve diet quality.

## 1. Introduction

Diet is known to be a vital component of daily life, as food provides essential energy, vitamins, minerals, and other nutrients for proper physiological function. From birth to the end of life, obtaining enough nutrition is necessary for growth, development, and health span [1]. Most of human history was punctuated with bouts of famine and food scarcity, making it difficult for humans to meet their caloric and nutritional needs. However, in most modernized countries, the industrialized food system has made calorically rich and nutrient-poor foods widely available, leading to a food environment that is conducive to energy toxicity [2,3]. This mismatch between high energy and low nutrient density in foods now making up a majority of food intake in Western countries is believed to be a major contributing factor to the uptick in the prevalence of chronic and metabolic disease [4]. In the U.S., for example, even with the inclusion of COVID-19, at least five of the top ten causes of death are diet-related [5,6,7]. Due to the significant role diet plays in the development of non-communicable diseases (NCDs), improving diet quality is paramount to enhancing public health.

While no specific diet is best for human health and disease prevention, many dietary patterns have been shown to decrease NCD risk, including the Dietary Approaches to Stop Hypertension (DASH), a Mediterranean diet [8], the Mediterranean-DASH Intervention for Neurodegenerative Delay (MIND) [9], NORDIC [10], and a whole-food, plant-based (WFPB) diet [11]. Based on the common aspects of these diets, a quality dietary pattern emphasizes fruits, vegetables, whole grains, legumes, and lean proteins while limiting excessive salt, added sugars, saturated fats, and highly processed foods. In addition to the types of foods present, the quantity of food is also important, as overconsumption is associated with NCDs as well [12].

While there is abundant evidence in the scientific literature of the components making up a healthy dietary pattern, health professionals and public health messaging, such as the United States Department of Agriculture (USDA) dietary guidelines, have made only modest improvements in public diet quality in the U.S. [8]. Individual dietary interventions by Registered Dietitians (RDs) are effective in improving diet and helping to manage diet-related NCDs [13,14]. However, this success is only seen in patients who are compliant with RD recommendations. Nutritional interventions suffer from poor patient compliance and high dropout rates [15]. The problem of patient compliance with dietary interventions is understandable due to the many barriers to changing diet quality. These barriers include factors that are difficult to overcome, such as environmental, social, and economic factors [16].

While unmodifiable barriers to improving diet exist, so do many lifestyle-related factors that individuals can leverage to indirectly lead to positive changes in diet quality. These lifestyle-related factors include exercise, adequate sleep, mindfulness practices, socialization at meals, social media use, and drug use, specifically alcohol and tobacco use. These factors were chosen as they are within the control of and modifiable by individuals. They are also accessible and do not incur financial burdens depending on the chosen modality.

Introducing these modifiable lifestyle-related factors as additional tools for RDs, physicians, and other health professionals providing dietary interventions may help increase adherence rates and lead to improved diet quality. Therefore, the purpose of this review is to examine the available evidence exploring the use of these lifestyle-related factors for improving diet quality and to provide health professionals with additional evidence-based diet interventions.

## 2. Methods

A narrative review was conducted to synthesize up-to-date findings from the literature to provide a summary of actionable interventions for healthcare professionals assisting patients with making dietary changes. Multiple databases were utilized to search for literature pertinent to this review, including PubMed, CINAHL, and Google Scholar. Search criteria included randomized controlled trials, reviews, systematic reviews, meta-analyses, and clinical trials. These studies were restricted to trials conducted on human subjects, published in English, and to literature published within the last ten years in peer-reviewed journals. Preference was given to studies published within the last five years to provide as much recent evidence as possible. Search terms included diet, diet quality, energy intake, caloric intake, exercise, sleep, mindfulness, mindful eating, intuitive eating, social life, socialization, social media, media use, tobacco, smoking, and alcohol. The literature included in this review was narrowed down first by title, then abstract, and finally by relevance. Additional literature was found while reviewing meta-analyses and reviews regarding the lifestyle-related factors discussed above.

## 3. The Impact of Lifestyle-Related Factors on Diet

### 3.1. Exercise

Exercise includes both aerobic and resistance training methods and has been defined as the expenditure of energy and the stimulation of skeletal musculature [17]. Exercise is known to increase metabolically active lean mass. This, along with the energy expended during exercise, has led to much of the research on exercise related to diet being focused on exercise as an intervention for counteracting excessive energy intake, thus reducing weight gain and potentially leading to weight loss. However, recent research has found that not only does exercise increase energy expenditure, but it may also impact food preference and appetite regulation [18].

Currently, there is limited research regarding the impact of exercise on food choice. A review of studies examining the effects of exercise on food preference showed that exercise reduces liking (r = −0.25, *p* < 0.001) and wanting (r = −0.27, *p* = 0.001) of high-fat foods [19]. High-fat foods are a good proxy for poor-quality foods, as many highly processed foods contain high levels of fat. Additionally, fat is the most calorically dense macronutrient, thus making foods high in fat more likely to lead to excessive energy intake. The reduction in preference for high-fat foods may be increased with greater exercise intensity and in individuals with greater adiposity [19]. These studies also showed increased liking and wanting of high-fat foods with increased sedentariness. These observations suggest that increasing exercise may be even more effective at increasing diet quality in sedentary and obese individuals [19]. Associations between increased physical activity and healthful food choices have also been shown in young adults and children [20]. In one study, a 15-week exercise intervention program showed a greater preference for a prudent dietary pattern (high in fruits and vegetables, low-fat foods, and low in fried foods and soft drinks) in young adults (β: 0.0623; 95% credible interval: 0.0159, 0.1111) [20]. Another study also found that children with higher levels of physical activity were shown to have a greater intake of fruits and vegetables (*p* < 0.001) and a lower intake of unhealthy foods such as soft drinks (*p* = 0.003) and savory snacks for girls (*p* = 0.011) [21]. More research is needed in this area as much of it is observational, thus making causality difficult to prove.

The mechanisms behind the potential effects of exercise on reducing preference for high-fat foods require further investigation as well. One potential mechanism is psychological. Reducing preference for high-fat foods may be due to the association of high-fat foods being unhealthy, therefore leading individuals who are exercising to pick lower-fat food options as they are trying to improve their health. Another explanation may be that exercise somehow buffers the liking and wanting of high-fat foods via modulating neuronal activity in the brain [19]. One such mechanism was shown via functional magnetic resonance imaging (fMRI) in individuals after a 60 min exercise session [22]. Their results demonstrated that brain regions associated with food reward showed decreased neuronal activity after exercise. For example, compared with the control group, neuronal responses to high-energy food were significantly reduced in the left and right insula (*p* < 0.005) and left middle occipital gyrus (*p* < 0.005). [22].

More evidence is available regarding the effects of exercise on appetite regulation and energy intake. Evidence from studies examining the effects of exercise on appetite regulation has shown effects across various stages of life, varying levels of adiposity, and in both metabolically healthy and unhealthy individuals [20,23,24,25,26,27].

In children and adolescents, exercise has been shown to reduce the quantity of food consumed after exercising [23]. Both intensity and timing of the exercise seem to be important for modulating the impacts of exercise on appetite regulation in children. High-intensity exercise has been shown to have a greater effect on regulating food intake in overweight/obese children compared to less-intensity exercise interventions (*p* < 0.05) [24]. Besides intensity, timing seems to be the other most important factor regarding the regulatory effects of exercise on appetite in younger age groups. Studies examining the effects of exercise performed before or after meals showed pre-meal exercise interventions to have a greater effect on reducing food intake at subsequent meals [28]. The effects of reduced intake of food at the first meal following exercise interventions did not influence feelings of appetite or intake of food at following meals [23]. This suggests that exercise may be most effective at modulating intake at the first meal following the exercise session.

In addition to the impacts on improving healthful dietary choices, the 15-week exercise intervention mentioned above also showed an increase in the ability to regulate appetite in young adults [20]. Similar results have been shown in non-obese adults as well [25]. Participants who engaged in low-, medium-, or high-intensity exercise habitually were given a high or low-energy preload (porridge) one hour before eating an ad libitum lunch. Both medium and high-intensity habitual exercise groups showed a greater reduction in food consumption at lunch after the high-calorie preload, whereas the low-intensity group did not (*p* < 0.01) [25]. This study shows that exercise positively impacts homeostatic appetite control in adults. Exercise has also been shown to be effective in reducing food intake in healthy obese/overweight adults (*p* < 0.01) [26] and in individuals with type 2 diabetes (*p* < 0.05) [27].

Exercise intensity seems to be an important factor when it comes to exercise’s effect on homeostatic appetite regulation. This effect has been observed in multiple studies showing that as the level of physical activity increases, so does homeostatic appetite regulation [18]. The relationship between increasing activity levels and increased homeostatic appetite regulation has been shown to follow a J-shaped curve, with sedentariness being associated with an inability to regulate appetite [18]. This suggests that adequate physical activity may be a prerequisite for optimum homeostatic appetite regulation.

The mechanisms behind the relationship between increased exercise and appetite regulation require further research. However, some studies suggest an effect on appetite-related hormones [26]. GLP-1, a hormone released by intestinal enterocytes during digestion that triggers insulin release and reduces hunger, is affected by exercise [26]. Research has shown that both basal and postprandial GLP-1 levels were increased in individuals who exercised regularly, and GLP-1 levels were found to increase further as the period of regular exercise increased from three to six months (*p* ≤ 0.04) [26]. This suggests that habitual exercise for extended periods may further increase appetite regulation. Additionally, ghrelin, a hormone that stimulates hunger, has been shown to decrease after acute exercise, while peptide YY (PYY), a pancreatic polypeptide hormone released during digestion that decreases appetite, has been found its secretion increased after exercise [18]. Consistent exercise may also increase sensitivity to endocrine signaling. Sensitivity to leptin and insulin, both of which have appetite-suppressing effects, has been shown to increase with consistent exercise [18]. Increased insulin sensitivity is thought to be induced through multiple mechanisms, including increased glucose transporter-4 (GLUT4) concentration, increased translocation of GLUT4 to cell membranes, increased capillarization of muscle tissue, and decreased intramuscular lipids [29]. These findings suggest that exercise improves homeostatic appetite regulation via endocrine signaling alterations [18].

### 3.2. Sleep

Sleep is another important modifiable lifestyle-related factor that can be leveraged to improve diet. Sleep has been shown to impact the types of foods and the quantity of calories consumed [30]. Two major factors affecting sleep quality are sleep duration and schedule consistency. Currently, adults are recommended to achieve seven to nine hours of sleep per night [31]. Sleep schedule consistency involves maintaining normal sleep and wake-up times.

Evidence regarding the impact of sleep quantity on diet has increased in recent years, showing the significant role adequate sleep plays in diet quality. According to Grandner and co-authors [32], compared with those who sleep longer, people who sleep less per night (five–six hours) consume less dietary fiber (*p* < 0.001), vitamin C (*p* < 0.001), lutein, and zeaxanthin (*p* = 0.011), which are indicators of diet quality because they are primarily found in fruits, vegetables, grains, nuts/seeds and legumes. Therefore, a decrease in these nutrients indicates a lower consumption of foods that are valuable for diet quality. Additionally, compared to normal sleepers (seven to nine hours), both longer sleepers and shorter sleepers were shown to have reduced variety in their diets (*p* < 0.001) [32]. Several studies have confirmed the effects of inadequate sleep on diet quality, finding reduced intakes of fruits, vegetables, beans, whole grains, and fish while increasing the intakes of confectionary foods [33]. Alcohol consumption may be affected by sleep duration as well. Both longer and shorter sleepers have been found to report increased alcohol intake above the recommended intake guidelines [32,34]. Individuals who sleep for longer durations have also been shown to be less likely to consume healthy dietary patterns [35].

The effects of inadequate sleep on diet quality may include both hormonal and psychological mechanisms. Experimental studies examining the impacts of sleep on hormones have shown that inadequate sleep leads to increases in ghrelin, which stimulates appetite [36,37,38]. Additionally, inadequate sleep was shown to decrease leptin, a hormone that decreases appetite [37,38]. However, other studies have found conflicting results, leading to a need for further studies on the hormonal effects of inadequate sleep [34]. Inadequate sleep may also negatively impact mood and lead to psychological distress. Due to the psychological distress induced by inadequate sleep, individuals may increase food intake, specifically energy-dense foods, to alleviate the distress [34].

Sleep consistency also affects diet quality, appetite regulation, glucose regulation, and energy metabolism [35]. Inconsistencies in sleep schedules, such as going to bed and waking up at different times, have been shown to distort circadian rhythms and dietary patterns. While some changes in sleep schedules may be difficult to avoid in such cases as shift work, constant work travel, or during early parenthood, other factors can be addressed. One such factor affecting sleep consistency is known as social jet lag, which usually involves individuals sleeping less during the week and sleeping for extended periods on weekends due to work and social factors. People may also shift their sleep schedules on the weekend, staying up later and then waking up later the following day [35]. Social jet lag was shown to be negatively associated with a healthy dietary pattern [35] (*p* < 0.006). Having a later sleep midpoint vs. an earlier sleep midpoint (waking up later vs. earlier) has been associated with skipping breakfast (*p* < 0.001) and eating later in the day [39].

Eating later in the day may be one of the mechanisms by which inconsistencies in sleep negatively affect diet quality and energy intake. Studies have shown that eating later in the day is associated with the consumption of energy-dense foods and results in decreased leptin levels, potentially leading to increased hunger (*p* < 0.0001) and, thus, food consumption on the subsequent day [40]. Studies assessing the association between meal timing and sleep quality have shown that people with later eating patterns (eating after 21:00) have lower sleep efficiency (total sleep time/time in bed) (mean [SE], 77% [2.3%]) [41]. Iao and colleagues analyzed the American Time Use Survey, which included more than 10,000 participants, and reported that waking during sleep occurred more than twice among people who ate or drank within an hour before bedtime (OR = 2.26, 95% CI: 1.93, 2.64) [42]. Additionally, only one night of fragmented sleep impacted appetite and could induce extended food consumption. This was due to insulin secretion being reduced the following morning and increased in the afternoon (*p* < 0.05). Concentrations and fullness scores of glucagon-like peptide 1 dropped as well (*p* < 0.05). [43]. Additionally, both skipping breakfast (*p* ≤ 0.001) and staying up later have been associated with increased binge behaviors [44]. This may be another mechanism by which sleep schedule inconsistencies affect diet quality and energy intake.

Energy intake has been shown to follow a similar trend to diet quality, with short sleepers consuming more calories per day compared to normal sleepers (RRR 1.024) [32]. Studies suggest that the increased intake of calories is due to increased food intake throughout the day and increased intakes of calorically dense foods [34]. One meta-analysis estimated that partial sleep deprivation leads to an increased intake of 385 calories per day (*p* < 0.00001) [31]. Another effect of inadequate sleep on caloric intake is the reduction in the ability to control appetite and binge impulses, thus leading to increased caloric intake [45]. Excessive sleep has also been shown to lead to an increase in caloric intake, with one study showing an estimated 500-calorie increase per day [35].

There are multiple potential mechanisms behind the effects of sleep duration on dietary energy intake. The simplest explanation may be that individuals who sleep less have more time to eat throughout the day. Studies have shown reduced sleep duration being associated with increased consumption of meals and snacks throughout the day [34,46]. Neuronal activity has also been shown to be affected by inadequate sleep, as during sleep deprivation, increased activity in the brain’s hedonic reward areas is present [47]. This may lead to an increase in food consumption due to the hedonic aspects of food overriding homeostatic hunger regulation in a sleep-deprived state [34]. Additionally, as mentioned above, inadequate sleep may increase the consumption of food to cope with increased psychological distress. Studies also suggest that individuals may increase their intake of energy-dense foods such as sweets due to fatigue during periods of inadequate sleep [34].

### 3.3. Mindfulness Practices

Mindfulness practices may be another lifestyle-related factor that can be leveraged to modify diet quality and energy intake. Multiple types of mindfulness practices have been used as interventions to improve diet. These interventions include mindfulness meditation, mindful eating, and intuitive eating practices. Mindful eating focuses on increasing awareness of hunger, fullness, taste awareness, and overeating triggers [48]. Similarly, Intuitive Eating emphasizes eating according to hunger and satiety cues, believing in your body to make the right decision [49].

While not directly aimed at affecting diet, mindfulness meditation and mindfulness practice have been shown to improve diet. In one study, researchers tested the effects of a 31-day web-based mindfulness meditation program on meditation-naive individuals with a tendency to stress eat [50]. This intervention was compared to a control that included informative health-based information via a web-based platform similar to the intervention group. Study participants reported a significant reduction in stress eating, emotional eating, and food cravings compared to the health training group [50]. Another study examined the effects of mindfulness training along with a dietary regimen compared to a mindfulness-only intervention and a dietary regimen intervention [51]. They showed that the mindfulness training intervention alone led to weight loss in obese individuals. This may have been due to a reduction in stress and emotional eating. While the mindfulness group did show improvement, the mindfulness and dietary regimen group showed greater improvements. This shows that mindfulness interventions may improve adherence to dietary interventions, leading to improved dietary quality and/or reduced energy intake. This study also showed that the combined mindfulness and dietary intervention had continued durability after two months and that mindfulness interventions may be effective over extended periods [51].

Mindfulness programs geared specifically toward eating have been developed and are known as mindfulness-based eating or mindfulness-based eating training. These programs include mindfulness practices that focus on increasing awareness of hunger, fullness, taste awareness, and overeating triggers [48]. Using mindfulness-based eating training, one study compared regular mindfulness training to mindfulness eating to examine the effects on eating behavior and binge eating [48]. While both interventions showed improvements in eating behavior and a reduction in binge eating, the mindfulness eating intervention showed greater benefits compared to the regular mindfulness training intervention (*p* ≤ 0.001) [48]. In another study, subjects participated in a diet and exercise program with one group also participating in mindfulness-eating training and found that compared to the control group, the mindful-eating group was able to maintain a reduction in eating sweets from 6 to 12 months while the control group significantly increased sweets consumption during the 6–12-month study period (*p* = 0.035) [52]. Although many studies show the beneficial effects of mindful eating interventions on diet quality and intake, there are also many studies that show no effect [53]. Of the many aspects of mindful eating interventions, those focusing on decentering and attention to the sensory properties of food seem to be most promising for improving diet [53].

Another mindfulness-related approach for improving diet is Intuitive Eating, which is similar to mindfulness eating interventions as it emphasizes becoming more attuned to physiological hunger and satiety cues and eating according to them [54]. However, Intuitive Eating also focuses on building a positive relationship between food and body and rejecting diet mentality [54]. Additionally, unlike other mindfulness practices that may affect diet indirectly, Intuitive Eating was developed to affect diet directly. Studies examining the effects of Intuitive Eating on diet behavior have found that those with higher Intuitive Eating scores have reductions in maladaptive eating behaviors such as emotional/reward eating and overeating [55]. Intuitive Eating has also been shown to be effective for disordered eating patterns that may lead to excessive food intake [54]. Of the disordered eating patterns studied, Intuitive Eating was shown to be most strongly effective at reducing binge eating (*p* < 0.01) [54]. This is important not only due to the reduced energy intake that is likely to occur if binging is reduced but also because the quality of the diet is important for individuals with binging tendencies. Evidence suggests that a majority of foods consumed during binging episodes may be comprised mainly of ultra-processed foods [56]. Therefore, Intuitive Eating may also improve diet quality to some extent. Intuitive Eating’s effects on diet quality are mixed in studies that do not specifically examine its effects on disordered eating patterns. Based on a recent systematic literature review, studies have shown a neutral or positive effect on diet quality [57]. This finding makes sense as Intuitive Eating is primarily focused on eating to physiological hunger cues and does not impose dietary restrictions. These findings suggest that those who practice Intuitive Eating consume calories that are closer to their physiological needs. This is backed up by evidence showing that those who practice Intuitive Eating have a more stable weight over extended periods of time [55]. The effectiveness of mindfulness practices on diet may be due to their effectiveness in reducing stress, specifically chronic stress, which has been shown to negatively impact brain areas that play a role in impulse control, hunger, and satiety [50]. For example, stress can negatively impact the hypothalamic–pituitary–adrenal (HPA) axis, affecting the production of hunger and satiety-related hormones like leptin, ghrelin, and neuropeptide Y [50]. Given the beneficial effects of mindfulness practices on stress, reducing stress via mindfulness may improve the function of brain regions affecting dietary intake. Indeed, studies examining the impacts of mindfulness practices have shown improvements in these brain regions and improved functional connectivity within the hypothalamus and insula, both of which play important roles in the neural processes mediating hunger and satiety cues [50]. Further research is needed to examine the mechanisms behind mindfulness practices on diet quality and intake.

### 3.4. Meal Socialization

Another strategy for improving diet may be to leverage social connections. One’s perceived social connectedness and who they choose to socialize with at meals may impact dietary choices and energy intake. Increasing social connectedness and collaborating with social connections to improve diet quality may be both effective and practical. Studies have shown that people who eat together tend to eat similar foods [58]. A study utilizing data collected from a hospital system showed that co-workers who ate lunch together tended to purchase similar foods. Additionally, this study showed that employees tended to purchase similar food items that their co-workers purchased at the previous meal. This association was shown for both healthy and unhealthy foods (*p* < 0.001) [58]. The same has been shown for the quantity consumed as well. With some exceptions noted above, individuals tend to mimic the amount of food eaten by the person they are dining with [59]. The exact number of shared meals required to make food choices more similar in meal participants was not specified in these studies.

Although the people we socialize with while eating can influence what we eat, a lack of social interaction can also influence what we eat. Those who perceive themselves as socially isolated have been shown to have a decrease in diet quality. In a study examining the effects of social isolation on eating behaviors following the COVID-19 pandemic, researchers showed that participants who scored higher for social isolation showed activity changes in specific brain regions [60]. The regions affected have been shown to play a role in food cravings, as well as areas involved in sensing appetite and hunger signals. This increased activity was especially pronounced when participants were shown images of sweet foods [60]. The increased reactivity to images of sweet foods in this study has been shown to translate to reported increased intakes in previous studies (*p* < 0.001) [61]. Other studies have shown a decreased intake of healthy foods with reduced socialization as well. Infrequent contact with friends and family has been shown to be associated with a reduction in diet variety, specifically in fruits and vegetables (*p* < 0.05) [62]. This same study showed that as the frequency of social contact increased, so did the intake of healthy foods [62]. Therefore, reduced social interaction may increase the consumption of unhealthy foods while decreasing the amount of healthy food intake.

In fact, everything from a person’s gender to their adiposity has been shown to influence how much the people they share a meal with will eat. Women have been shown to eat more when sharing a meal with another woman and less when sharing a meal with a male, while men have been shown to eat more when sharing a meal with a woman [63]. Perceived adiposity of an eating partner has also been shown to affect food intake. Experimental studies pairing participants with either normal-weight or overweight confederates have shown that when paired with an overweight confederate, participants ate less compared to when eating with a normal-weight confederate [59]. The number of people one eats with also has been shown to affect food intake. Studies have shown that when eating in large groups, people tend to eat more, but one exception to this is when women eat in mixed-gender groups [63]. While these social influences may be useful for people to know when trying to improve their diet, changing who one eats with based on these findings may not be practical.

Under different circumstances, social gatherings may affect people’s diets in adverse ways. For example, alcohol has played an important role in human history and social interactions. A significant positive association was found between alcohol consumption and the number of forks present (people sharing a meal) (*p* < 0.05) [64]. Alcohol ingestion is also heavily influenced by culture. Countries have been classified into different categories according to the quantity of alcohol consumed [65]. Category 1 includes Mediterranean countries. These diets contain moderate amounts of red wine or beer. Australia and Japan have also been classified into this category. Higher category numbers indicate a higher risk of diseases and mortality. The United States and Canada fit in the second category, and South America, Africa, Southeast Asia, and Russia belong to the third category. In Central American regions such as Guatemala and Nicaragua, the risk of morbidity and mortality is highest due to their heavy drinking patterns [65]. Social behavior could be influenced by health education. Therefore, more efforts should be made to increase awareness of the positive aspects of socialization while encouraging a reduction in the role alcohol plays. This is especially important for countries in the higher categories.

### 3.5. Social Media Use

Social media use includes both the platform and the amount of time spent on the platform. Social media is a novel influence on individuals’ dietary choices that RDs and other health professionals have not had to contend with until relatively recently. Both social media influencers with strong contrarian nutritional claims and food advertisements on social media platforms can interfere and contend with dietary recommendations from reputable health organizations and the USDA Dietary Guideline for Americans [66]. Given social media’s potential effects on dietary choices, modifying its use may be another factor health professionals can try to indirectly affect diet quality.

The effects of social media use on adults seem to be variable, with the effects being positive or negative depending on the content that the adult is exposed to and the time spent on social media [67]. In one study, 51% of participants described changing their diet due to social media content, and 71% reported positive dietary changes. However, as social media use increased, reported social media benefits on dietary choices decreased [67]. This study, however, involved many participants who were nutrition students, which may have impacted the content viewed and led to more beneficial effects. A recent meta-analysis showed that users were more likely to consume foods seen on social media [68]. However, this association was not present when studies examined the effects of healthy food content on eating behaviors [68]. Social media may also affect diet quality via food restriction. Studies have shown that social media may reduce diet quality by leading to dietary restrictions [69]. This may decrease the quantity of healthy foods consumed by users, thus reducing their diet quality. Although restriction of food intake may be beneficial in individuals who over-consume calories, these studies indicate social media use increases over restriction, leading to increased rates of disordered eating [68,69]. While more research is needed to determine the impacts of social media on diet quality in adults not related to disordered eating, abundant studies showed associations between increased social media use and the development of eating disorders in young adults and children [70]. This is most concerning as eating disorders are very detrimental to health. Anorexia nervosa, for example, has the highest mortality rate among psychiatric disorders [71]. Therefore, RDs and health professionals should be very vigilant when speaking to patients about social media use in the context of diet.

Research on the influence of social media on diet also showed a significant impact on the diet of adolescents and children. As with adults, excessive social media use has been shown to have negative effects on diet quality in adolescents. Adolescents exceeding media time recommendations (<2 h) were shown to have lower diet quality scores (*p* = 0.002) [72]. Additionally, this study showed that children who used more social media had lower intakes of nutrients that are important to growing children, such as vitamin D and calcium. Studies have shown that when exposed to influencers and advertisements pushing unhealthy snacks, children increase their consumption of unhealthy snacks, leading to increased caloric intake as well [73].

These social media food advertising tactics are especially effective in younger children as they have yet to fully develop the brain’s frontal lobe, which is critical for the executive functioning required for inhibitory control and blunting the effectiveness of food advertisements [73]. Video-based platforms such as YouTube and TikTok may be the most influential in affecting children’s diet quality. These platforms have been shown to contain the most food advertisements per post for children, with YouTube ranking the highest. The most advertised foods on these platforms were fast food, soft drinks, snacks, candy and chocolate, and water [74]. This is concerning because, with the exception of water, these foods are energy-dense and nutrient-poor.

### 3.6. Tobacco and Alcohol Use

Two of the most common drugs used by adults are tobacco and alcohol. Both have been shown to be associated with reduced diet quality. In one study, researchers utilized the National Health and Nutrition Examination Survey (NHANES) data to examine the association between diet quality using the Healthy Eating Index (HEI) and smoking status [75]. In this study, researchers found statistically significant associations between diet quality and smoking status. In addition, both new and former smokers scored higher than current smokers on the HEI (*p* < 0.001). HEI scores showed further decreases as the level of smoking increased, indicating a negative association between diet quality and the number of cigarettes smoked. Furthermore, smokers were found to have an increased caloric intake and a preference for energy-dense or empty-calorie foods. However, this study also found that upon smoking cessation, HEI scores showed immediate improvement, nearly equivalent to those of non-smokers within five to ten years (*p* > 0.05) [75]. The associations between smoking status and reduced diet quality have been demonstrated in studies conducted in Europe and Asia as well [76,77]. Smoking cessation may be especially important in the context of improving diet quality for the prevention of non-communicable diseases, as smoking has been shown to negate the impact of a high-quality diet on disease prevention [78].

The impact of smoking status on diet quality may be multifactorial. Studies have shown that smoking may impair or change taste perceptions of food, leading to preferences for foods of poor nutrient quality [79]. However, other factors may play a role, as smokers are more likely to be less educated and of poorer socioeconomic status, both of which may also contribute to poorer diet quality [79].

Similarly, alcohol consumption is associated with reduced diet quality. For example, one study conducted in Spain found that alcohol consumption was associated with lower rates of adherence to the Mediterranean pattern of eating (OR 1.35 low-risk drinkers and 1.54 high-risk drinkers) [80]. A meta-analysis of the literature found similar findings that alcohol consumption appears to increase the desire for fatty and protein-rich foods and decrease cravings for sweet foods [81]. Another study conducted in eight Latin American countries found that as alcohol consumption increased, diet quality decreased (*p* = 0.001) [82].

This study also found that total caloric intake tended to increase with alcohol consumption as well (*p* = 0.001). While total caloric intake increased, energy intake from all macronutrients decreased in alcohol consumers, with alcohol comprising nearly 17% of caloric intake (*p* = 0.001). In a randomized controlled trial (RCT), participants who consumed an alcoholic beverage prior to an ad libitum meal expressed an increased desire for high-fat savory foods and consumed, on average, 11% more energy compared to the control group (*p* = 0.004) [83]. In terms of increasing consumption, research seems to support a dose-dependent response. Food intake tends to increase during episodes of low to medium levels of alcohol consumption, while caloric intake may increase to a greater extent during heavy drinking episodes; this seems to be due to the caloric content of the alcohol consumed [84]. More research is needed to solidify the relationship between alcohol intake and diet quality, as much of the research is observational.

The mechanisms behind alcohol’s influence on diet may be related to dopamine, as at lower doses, alcohol has been shown to stimulate dopamine release in the brain, which may, in turn, lead to seeking out rewarding foods [81]. At higher doses, alcohol may lead to excessive dopamine in the neural pathways, leading to decreased motivation. In addition, alcohol’s effects on mood may also affect food intake while drinking since, at low doses, the mood is elevated, thereby increasing motivation. Whereas at high doses, alcohol has sedative-like effects, decreasing motivation. Lastly, at high doses, alcohol consumption may lead to increased satiety, causing decreased hunger as it fills the stomach [81].

It is also important to note that alcohol consumption may not only impact diet quality but has also been shown to impact nutrient absorption [85] and liver health. Heavy doses (blood alcohol content ≥ 0.08) have been shown to impair amino acids, B vitamins, fat-soluble vitamins, and mineral absorption, among others [85]. Unrestricted alcohol consumption could also damage the liver and result in alcoholic liver disease (ALD), including fatty liver, alcoholic hepatitis, liver fibrosis, cirrhosis, and liver cancer. This damage is due to the injury of tissues and organs as a consequence of the inflammatory cascade from a multitude of cytokines, chemokines, and reactive oxygen species due to the harmful metabolites generated by alcohol metabolism [86]. Primary and secondary protein/energy malnutrition is often present in ALD patients. The mechanisms of malnutrition include reduced dietary intake because of the replacement of food by alcohol consumption, insufficient appetite due to dysgeusia and inflammation in the gastrointestinal tract, malabsorption due to exogenous pancreatic insufficiency and diarrhea, and low palatability of food with inclined sodium. [87].

The different effects of lifestyle-related factors on diet quality and energy intake are summarized in Table 1. References show the sources from which information was obtained.

## 4. Limitations

Although this review is profound, some limitations should be recognized. First, this review was confined to investigations published in English, which might have expelled corresponding discoveries from non-English-speaking territories. Also, people’s diets could be influenced by many factors, including culture and religious beliefs. Additionally, observational and survey data for many of the sections were incorporated in this review, so this paper relied on a fair number of correlational findings. Furthermore, a number of research examining factors related to diet quality and lifestyle are descriptive studies with relatively small sample sizes. More investigations with a larger number of participants are needed. However, we have included relatively comprehensive information, including review articles and most recently published papers.

## 5. Conclusions

Based on the findings in this review, diet is not a simple choice. Dietary patterns are influenced by a complex interplay of genetic, societal, environmental, economic, and lifestyle factors. While many of these are unable to be controlled or modified, the lifestyle-related factors discussed above may offer additional points of intervention for RDs and other health professionals to improve the diet quality of their clients. Health professionals can suggest the following: consistent exercise (ideally of moderate to vigorous intensity) to increase diet quality and improve homeostatic appetite control, a consistent sleep schedule that ensures proper sleep duration, the use of one of the various mindfulness modalities to enhance the quality of diet and promote reconnection with physiological hunger cues, the use or growth of social connections to assist in improving diet quality, caution against excessive use of social media (especially for pediatric patients) to reduce exposure to unreliable nutrition misinformation and exposure to the commercial influence of low-quality food products, and the reduction or cessation of tobacco and alcohol use. While not all the lifestyle-related factors discussed above may be practical to modify for everyone, health professionals can offer education regarding how each may impact a person’s diet and allow them to identify which, if any, of the factors may be beneficial to modify for them. By educating people on how these factors may affect diet quality and leveraging factors that are practical for a person to modify in their life, chances of successfully improving diet quality may increase.

## Figures and Tables

**Table 1 nutrients-17-00448-t001:** Effects of Lifestyle Related Factors on Diet Quality and Energy Intake.

Factors	Impact on Diet Quality	Impact on Energy Intake	Reference(s)	
Adequate Exercise	+	−	[19,20,21]	[20,23,24,25,26,27]
Good Sleep Quality	+	−	[32,33,34,35,42]	[31,32,34,35,39,43,44]
Practicing Mindfulness	+	−	[47,48,49,52,53,54]	[45,47,48,52,53]
Social Connection	+	+/−	[59,60]	
Excessive Social Media Use	−	+	[62,63,64,65,67,69]	[68]
Tabaco Use	−	+	[70,71,72]	[70]
Alcohol Consumption	−	+	[75,76,77]	[76,77]

Note: “+” means increase, “−” means decrease, and “+/−” means inconsistent and it depends on the situation.

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
