# Peer review of "Lifestyle-Related Factors for Improving Diet Quality"

_nutrients, 2025, doi:10.3390/nu17030448_

Round 1

Reviewer 1 Report

Comments and Suggestions for Authors

This literature review emphasizes the importance of different lifestyle factors to modify the dietary pattern in the population. It is interesting the relationship found between these factors and appetite hormones (sleep and ghrelin or leptin or with caloric intake, excessive and deprivation increase the energy intake in 400-500 cal/day). In addition, it was motivating to read the gender difference in socialization in meals. Some recommendations to increase the understanding of the report would be:

- All section could be summarized in a graphical plot or combination between figures and tables.

- The sections could benefice to add data from the original manuscript, such as mean differences, OR, RR.

- Lines 128-129. It is not clear what is the food reward mechanism and what pretend to say.

- What can the diet interventions be to improve each factor? For example, would advise dairy products or relax-infusion to resolve sleep? Could implement the sections.

- Limitations. Following the English-language limitation, it would implement the social context, particularly in important for section 3.4, Latin-context can spend long time eating or even cooking.

Minor comments:

- Line 44. Define NCDs

- Section 3.4. How many times should be adequate to perform meal socially?

Author Response

Responses to Reviewer 1:

Comments 1: All section could be summarized in a graphical plot or combination between figures and tables.

Response 1: Thank you for your advice! It's a good idea to use a table or chart to summarize the impact of lifestyle factors on diet quality. We agree with this comment and have generated a table that can be found in the revised manuscript in the penultimate paragraph on page 13, lines 559-563.

Comments 2: The sections could benefice to add data from the original manuscript, such as mean differences, OR, RR.(add OR/RR for ref. That have them)

Response 2: Thank you for pointing this out! We agree with this comment and added some data from the original papers to the manuscript. They were highlighted in red in the revised manuscript and could be found in many places, such as on page 3, the last paragraph, lines 111, 123-124, on page 4, the first paragraph, lines 125-127, etc.

Comments 3: Lines 128-129. It is not clear what is the food reward mechanism and what pretend to say.

Response 3: Thank you for point it out! We apologize that it was not clearly explained in our original manuscript. What we want to say was the brain regions associated with food reward include the orbitofrontal cortex (OFC), amygdala, insula, striatum, and hypothalamus. While the neuronal response to high-energy food vs. control food has been found modified by exercise on brain regions, such as insula, gyrus, etc. This could explain people's food preferences changed after exercise. We have added some information to give an example for readers to understand. It could be found on page 4 of the revised manuscript, second paragraph, line 138-140.

Comments 4: What can the diet interventions be to improve each factor? For example, would advise dairy products or relax-infusion to resolve sleep?

Response 4: Thanks for your questions! you are right. Diet may in turn affect those lifestyle factors, for example certain foods may improve sleep quality or exercise performance. However, we are trying to focus on how lifestyle factors influence eating habits. Therefore, in this manuscript we are unable to discuss the impact of diet on these lifestyle factors as this would significantly increase the length of this review.

Comments 5: Limitations. Following the English-language limitation, it would implement the social context, particularly in important for section 3.4, Latin-context can spend long time eating or even cooking.

Response 5: Thank you for your advice! We agree with this comment and add that information about cultural and religious beliefs may also influence people's diets, which can be found in the last paragraph of page 13, lines 567-568.

Comments 6: Line 44. Define NCDs

Response 6: Thank you for pointing this out! We agree with this comment and defined the NCDs on page 2, first paragraph, line 44. 

Comments 7: Section 3.4. How many times should be adequate to perform meal socially? (how many times eating in a social setting is adequate to maintain or improve diet quality. Look at ref. 57-60)

Response 7: Thank you for your advice! These studies did not specify the exact number of meals that would need to be shared in order for meal participants to have more similar food choices. This information has been added to page 9, second paragraph, lines 377-378 and the same page, third paragraph, lines 392-393, as well as page 10, first paragraph, lines 409-410.

Reviewer 2 Report

Comments and Suggestions for Authors

Dear Authors,

The idea of this review is excellent and presents great potential as a source of information for health professionals. However, some ideas are mixed together, and the text can be confusing at times. Our recommendations will primarily focus on addressing this issue.

Keywords, please remove diet quality because it is in the title.

Line 44, For the first mention of "NCDs" in the text, we suggest writing it out in full and removing "these" from the sentence.

Lines 71 and 72, please justify how did you choose those lifestyles related factors and include references for it. It is important to justify because it is the basis of your terms search and your review discussion.

Line 81, maybe it is of interest to justify witch kind of review you used (It look like a narrative review?)

Line 82, in search criteria we suggest add peer reviewed journals

Point 3. We suggest in all the subchapters following a structure to avoid mixing the concepts, it will help you to be more synthetic and avoid repetitions and will be easier to understand and interpret for future lecturers. For example:  Exercise influence on Food choice following by mechanism of influence on food choice, until finishing the topic for all the kind of population involved. After influence on energy intake and appetite regulation, mechanism of influence on energy intake and appetite regulation until finishing the topic for all the kind of populations involved. After doing the same for Sleep, ….

Lines 94 to 96, they are many other reasons for exercise improving health, anti-inflammatory, anti-oxidant effects, ... Or you justify your choice in the present review or you should be more generalist in your presentation with some more references.

Lines 104 to 111, please check the text to avoid some confusion about witch effect? (reduce preference or increase preference?). Line 109 these studies, which one?

Please check lines 142 to 145, to avoid some confusion.

Lines 149 to 153, to avoid confusion, please clarify what is preload (before exercise?), Lunch (before or after the exercise?).

Lines 154 to 161, we suggest reducing this part, as it only complements information already provided. For example: "Six months of exercise also reduced food intake in metabolically unhealthy individuals, such as … (references)."

Line 211, “fruits, vegetables, oily fish, nuts and seeds” is a repetition we suggest remove and only let the new information’s.  

Lines 217 and 218: “which stimulate appetite” It is the first time you present this effect of sleep and it is in the mechanism to explain the effect. First explain all the effect of sleep, after explain the mechanisms as we suggested for point 3. It is important to follow a structure avoiding mixing the concepts.

Lines 259 to 265, Eating later also may affect the sleep quality that after will affect diet quality? Maybe it is of interest to discuss.

Meal socialization. what about alcohol consumption and cultural influence? Maybe it is of interest to discuss.

Sentence of lines 462 and 463, what happen to non-alcoholic calories consumption?

About alcohol consumption mechanism impact we suggest discuss how Impact of alcohol on liver health may decrease diet quality.

To show how your review findings, may help health professionals be able to educate people about lifestyle-related factors that can improve diet quality, we recommend a chapter 6. Practical Applications. Highlighting the key practical findings of this review to provide actionable insights for improving diet quality among individuals.

Author Response

Responses to Reviewer 2:

Comments 1: Keywords, please remove diet quality because it is in the title.

Response 1: Thank you for point it out! “Diet quality” has been deleted from the keywords. Please see the bottom of page 1, line 28.

Comments 2: Line 44, For the first mention of "NCDs" in the text, we suggest writing it out in full and removing "these" from the sentence.

Response 2: Thank you for pointing this out! We agree with this comment and defined the NCDs on page 2, first paragraph, line 44.  “These” has been removed from the sentence.

Comments 3: Lines 71 and 72, please justify how did you choose those lifestyles related factors and include references for it. It is important to justify because it is the basis of your terms search and your review discussion.

Responses 3: Thank you for pointing it out! These factors are listed as modifiable risk factors for many illnesses/chronic conditions however there is no agreed upon definition for what is considered a modifiable risk factor is. https://doi.org/10.1371/journal.pgph.0002887. Additionally, these factors were noted to influence diet during the author’s academic and clinical experiences which also played a role in these lifestyle factors being topics of interest for the review. The explanation about this has been added on page 2, penultimate paragraph, lines 73-75.

Comments 4: Line 81, maybe it is of interest to justify witch which kind of review you used (It look like a narrative review?

Response 4: Thank you for pointing it out! Yes. It is a narrative review and it has been added to page 3, second paragraph, line 84.

Comments 5: Line 82, in search criteria we suggest add peer reviewed journals.

Response 5: Yes, it is good to add “peer reviewed journals”. Please see page 3, second paragraph, line 91.

Comments 6: Point 3. We suggest in all the subchapters following a structure to avoid mixing the concepts, it will help you to be more synthetic and avoid repetitions and will be easier to understand and interpret for future lecturers. For example: Exercise influence on Food choice following by mechanism of influence on food choice, until finishing the topic for all the kind of population involved. After influence on energy intake and appetite regulation, mechanism of influence on energy intake and appetite regulation until finishing the topic for all the kind of populations involved. After doing the same for Sleep, …

Response 6: Thank you for pointing them out! We agree that explaining the mechanism following the different impacts will make it more logic and clear. Please see page 6, the second to last paragraph and the third to last paragraph (highlighted in yellow avoiding cover other highlights), lines 232-262, as well as page 8, the last paragraph, lines 337 to page 9, first paragraph, line 364.

Comments 7: Lines 94 to 96, they are many other reasons for exercise improving health, anti-inflammatory, anti-oxidant effects, ... Or you justify your choice in the present review or you should be more generalist in your presentation with some more references.

Response 7: Thank you for your advice! Yes, exercise improves health through different mechanisms. We didn’t discuss that because we tried to focus on how lifestyle factors impact diet.

Comments 8: Lines 104 to 111, please check the text to avoid some confusion about which effect? (reduce preference or increase preference?). Line 109 these studies, which one?

Response 8: Thank you for your suggestion! The paragraph has been rewritten. “The reduction in preference for high-fat foods may be increased with greater exercise intensity and in individuals with greater adiposity [19]. These studies also showed increased liking and wanting of high-fat foods with increased sedentariness.”. Please see last paragraph on page 3, lines 112-119. “These studies” refers to the studies reported in reference 19.

Comments 9: Please check lines 142 to 145, to avoid some confusion.

Response 9: Thank you for your suggestion! The paragraph has been rewritten. Please see page 4, penultimate paragraph, lines 151-155.

Comments 10: Lines 149 to 153, to avoid confusion, please clarify what is preload (before exercise?), Lunch (before or after the exercise?

Response 10: Yes, preload means providing some porrage before an ad libitum lunch. There is no exercise around the meal time. Participants were assigned into different groups based on their low-, medium-, or high-intensity exercise habit. Please see page 4, last paragraph, lines 161-163.

Comments 11: Lines 154 to 161, we suggest reducing this part, as it only complements information already provided. For example: "Six months of exercise also reduced food intake in metabolically unhealthy individuals, such as … (references).

Response 11: Thank you for pointing it out! We agree with it and revised the writing to make it concise. Please see page 4, last paragraph, lines 166-168.

Comments 12: Line 211, “fruits, vegetables, oily fish, nuts and seeds” is a repetition we suggest remove and only let the new information’s.

Response 12: Thanks for the suggestions! We agree with it and the repeated part has been deleted. Please see page 6, second paragraph, line 222-224.

Comments 13: Lines 217 and 218: “which stimulate appetite” It is the first time you present this effect of sleep and it is in the mechanism to explain the effect. First explain all the effect of sleep, after explain the mechanisms as we suggested for point 3. It is important to follow a structure avoiding mixing the concepts.

Response 13: Thank you for pointing it out! Good catch! The paragraphs have been rearranged. We discussed the impact of sleep on appetite and then sleep on diet quality and energy intake. Please see page 6, second and third paragraphs, lines 221-244.

Comments 14: Lines 259 to 265, Eating later also may affect the sleep quality that after will affect diet quality? Maybe it is of interest to discuss.

Response 14: Thanks for pointing it out! We agree with it. We added more information and references related to late eating reduced sleep quality, and then impacted the following day’s diet quality. Please see page 6, penultimate paragraph, lines 246-262

Comments 15: Meal socialization. what about alcohol consumption and cultural influence? Maybe it is of interest to discuss.

Response 15: We agree with it. The information has been added to the manuscript. Please see page 10, third paragraph, lines 426-441.

Comments 16: Sentence of lines 462 and 463, what happen to non-alcoholic calories consumption?

Response 16: While total caloric intake increased, energy intake from all macronutrients decreased in alcohol consumers with alcohol ingestion. The information has been added to the manuscript. Please see page 12, fourth paragraph, lines 524-526.

Comments 17: About alcohol consumption mechanism impact we suggest discuss how Impact of alcohol on liver health may decrease diet quality.

Response 17: Thank you for the suggestion! We agree that it is important to discuss the alcohol impact on liver health which further reduce the diet quality. The information has been added to the manuscript. Please see page 13, the first paragraph, lines 545-558.

Comments 18: To show how your review findings, may help health professionals be able to educate people about lifestyle-related factors that can improve diet quality, we recommend a chapter 6. Practical Applications. Highlighting the key practical findings of this review to provide actionable insights for improving diet quality among individuals.

Response 18: Thanks for pointing it out! We agree that readers will understand the main ideas of this article and gain information useful for practice. The findings have been summarized, but not enough to warrant another chapter. So, we put it in the conclusion section. Please see page 14, second paragraph, lines 581-590.

Round 2

Reviewer 2 Report

Comments and Suggestions for Authors

Dear Authors,

thank you for your effort, in responding to all our recommendations.

Great job!

Wishes of success with this publication